# Conceptual Assessment of the Possibility of Using Cryogenic Fuel on Unmanned Aerial Vehicles

**Anatolii Kretov [1], Vyacheslav Glukhov [2] and Alexey Tikhonov [3,*]**

[1]  College of Aerospace Engineering, Nanjing University of Aeronautics and Astronautics, Nanjing 210016, China

[2]  Gorbunov Kazan Aviation Plant, Branch of Public Joint-Stock Company Tupolev, 420036 Kazan, Russia

[3]  Moscow Aviation Institute, National Research University, 125993 Moscow, Russia

[*]  Correspondence: tikhonovmai@mail.ru

**Abstract:** The study is devoted to the creation of modern unmanned aerial vehicles (UAV), the most efficient from economic and environmental points of view. In connection with the vital need to switch to environmentally friendly vehicles, this research analyzes the possibility of using a cryogenic fuel (CF) for UAVs with piston or gas turbine engines. The numerical studies' analyses of the takeoff weight of the currently widely used UAV MQ-9 Reaper show the practical impossibility of using liquefied hydrogen and the low efficiency of using liquefied natural gas (LNG) as fuel for similar UAV with takeoff weight up to 5 tons.

**Keywords:** conceptual design; unmanned aerial vehicles (UAV); cryogenic fuel (CF); kerosene; liquefied natural gas (LNG); methane; hydrogen; takeoff mass sensitivity analysis to design changes

## 1. Introduction

UAV is a special type of aircraft, the main feature of which is the absence on board of the most invaluable yet the weakest "element" in the entire aircraft system, a person, with all the advantages and disadvantages that follow from this. The advantages of UAV are its ability to fly, unlimited time and overload, a lack of influence from the human factor that can lead to errors, reduction in the mass of the service load and the volume required for its placement, and no need for sealed compartments of a large volume.

UAVs have been incorporated into all spheres of human activity for a long time. The world's first aircraft, the Montgolfier brothers' balloon, was launched in 1783, climbed to a height of 500 m, flew for eight minutes, and successfully landed two miles from the launch site, essentially becoming the first UAV. Unfortunately, many great discoveries and achievements began their practical application initially for military purposes. The concept of the military application of UAVs dates back to 1849, when Austria attacked Venice with the help of unmanned balloons filled with explosives. The Austrian troops besieging Venice launched about 200 of these incendiary balloons over the city. In 1858, a balloon was first used for military aerial photography.

Gradually, UAVs for practical use were implemented in applications in addition to the aerostatic principle of aircraft flight. In 1896, A. Nobel launched a rocket with a camera. In 1933, based on the Fairy Queen biplane in the United Kingdom, the first radio-controlled reusable UAV, called the DH.82B Queen Bee, was created, which was used as a training target to simulate air targets. The German V-1 flying bomb became the first cruise UAV with a jet engine and an autonomous control system. Its launch was carried out strictly in the firing plane using a 50 m-long steam catapult. The autopilot kept the aircraft on a given course and altitude using gyroscopes and a magnetic compass. The corresponding range was set on the mechanical counter before the start, and the vane anemometer, placed on the nose of the UAV and rotated by the oncoming airflow, twisted the counter to zero when the required range was reached. At that moment, the fuses of the warhead were cocked,

and a dive command was issued. The accuracy of hitting the target of the V-1 rocket was $\pm6$ km. In total, about 25,000 units of this deadly type of weapon were produced, and about 20,000 sorties were made.

A breakthrough in the history of the development of drones, but again for military purposes, was in 1978 when the Scout UAV for the purposes of surveillance and reconnaissance was developed by the Israeli company IAI in 1978. The success of the Israeli UAV program in the early 1980s made it clear that these vehicles would play an increasingly important role on future battlefields. In 1986, the USA and Israel jointly developed the AAI RQ-2 Pioneer, which today is considered one of the basic platforms for UAVs. At that time, a new term began to be used for such aircraft: the drone.

UAVs have many elements that are different from manned air vehicles. They are usually classified according to their capabilities and size to carry out their respective mission. It is obvious that the goal set in the paper to assess the use of cryogenic fuel (CF) can be realized only in relation to large UAVs. Without going into the details of all the capabilities and classification of UAVs, we will focus only on such UAVs that can be redesigned to use CF and solve exclusively civilian tasks. We use the classification of military UAVs as follows:

- UAV of High-Altitude Long Endurance (HALE) type, capable of flying at altitudes of more than 15 km, flight duration more than 24 h;
- UAV of Medium-Altitude Long Endurance (MALE) type, designed for flight altitudes of 5–15 km, flight duration up to 24 h. Such UAVs usually operate at ranges of about 500 km;
- UAV of Medium-Range or Tactical UAV (TUAV) type, tactical UAV with a range of about 100–300 km. These aircraft are smaller and operate in simpler systems than HALE or MALE.

From the analysis of the layout schemes of today's UAV, two main categories can be distinguished: the normal model and the integration scheme of the wing with the fuselage: Blended Wing Body (BWB). Furthermore, for the time being, we will focus only on UAVs of a normal aerodynamic configuration (fuselage, wing located in the region of the center of mass of the aircraft, and plumage in the tail section). Currently, one of the most famous such UAVs is the MQ-1 Predator, which, in addition to reconnaissance missions, is also capable of destroying targets. In 2007, another strike UAV appeared: the MQ-9 Reaper. To date, the largest of the most used UAV is the RQ-4 Global Hawk, which first took to the air in 1988. Of the latest Russian UAV, the Altius and Helios can be noted.

The main characteristics of the listed UAV are given in Table 1 (due to incomplete information about these devices, some parameters are missing) [1].

The relevance of the use of civilian UAVs is obvious. Some of the tasks they can successfully perform are aerial photography and video shooting, monitoring of the environment and sources of pollution, oil and gas pipelines and power lines, farmland, fires, coastlines, crop processing, search and rescue of people in extreme and emergency situations, customs supervision of illegal imports, use in fishing to search for schools of fish, pilotage of ships on the northern sea route, delivery of goods and mail, meteorological observations, traffic control and management, local scale mapping, etc.

**Table 1.** Main characteristics of some modern UAVs.

| UAV | Length, m | Wing Span, m | Empty mass, kg | Payload, kg | MTOM, t | Powerplant | Max. Speed, km/h | Cruise Speed, km/h | Endurance, h | Range, km |
|---|---|---|---|---|---|---|---|---|---|---|
| MQ-1 Predator | 8.2 | 14.8 | 512 | 0.3 | 1.02 | 1 × Rotax 914F86 kW | 217 | 130–165 | 20–40 | 726 |
| MQ-9 Reaper | 11 | 21.3 | 2.223 | 1.760 | 4.760 | Honeywell TPE331-10T712 kW | 482 | 276–313 | 14–30 | 5926 |
| RQ-4 Global Hawk | 13.5 | 35.4 | 3850 | – | 10.400(15) | 1 × Allison R-R AE3007H31.4 kN | – | 650 | 36 | 22,000 |
| Helios | 12.6 | 30 | – | 0.8 | 4 | – | – | 450 | ~30 | 3000 |
| Altius | 11.6 | 28.5 | – | 1–2 | 5–7 | 2 turbocharged diesel RED-A03/V12 500 hp (680 kW) | – | 150–250 | ~48 | 10,000 |

Drones have been seriously considered an effective means of transporting cargo or people for the last 10–15 years. In any case, research is being carried out to find the optimal design and technological solutions. By optimal UAV solutions, we mean a set of characteristics that UAV can provide as a cost-effective, safe, and reliable mode of transport. In recent years, an important feature of environmental safety has been added to these characteristics, i.e., minimal impact on the environment [2]. To systematize this approach, the level of the so-called "carbon footprint" is taken, which is formed during the production of the vehicle, its maintenance, and operation. For aviation equipment, operating costs are a very significant cost item. For example, for passenger airlines, the cost of kerosene is 20–30% of the cost of air transportation [3]. Unlike manned aircraft, UAVs do not require significant expenses for the training and payroll of the flight crew, as they are controlled automatically by an operator on the ground. Thus, the share of UAV fuel costs will be higher than in manned aviation, and the task of switching to more economical and environmentally friendly fuel becomes even more urgent. The following requirements are imposed on promising aviation fuels for UAVs: high specific heat of combustion, low carbon footprint, acceptable cost, etc. Minimum carbon footprint requirements should be clarified. According to accepted international concepts regarding fuels, in calculating the level of the "carbon footprint", it is required to consider the level of $CO_2$ during combustion and during its production [4]. These requirements are best met by hydrogen and natural gas. An important requirement for aerodynamic aircraft is the minimum aerodynamic drag. This condition, accordingly, forces a reduction in the volume of fuel tanks, which in turn requires the use of fuel with a maximum specific gravity, so promising fuels for UAVs must be on board in a liquid (cryogenic) state.

## 2. Literature Review

The issue of justifying the use of CF for UAVs is fundamental, and it should be considered at the very beginning. Given that the designer has the very minimum of initial data at this stage, the methodology for such justification should be quite simple but reliable and convincing. Existing works in this direction [5–8], as a rule, are based on solid software products—they require extensive preliminary research and a deep study of possible options.

Currently, most experts are inclined to conclude that the use of CF in UAVs should go exclusively in the direction of fuel cells [9,10], that is, using electric traction. However, considering the accumulated experience of creating a flying laboratory based on the Tupolev Tu-154 aircraft with engines operating on CF [11,12], the authors set out to carry out a weight assessment of the use of CF for UAVs using gas turbines or piston engines.

According to many authoritative experts, using hydrogen as a fuel is the main factor in developing the entire strategy for future aviation, including unmanned aircraft. However, it is rather difficult to make such a transition in one step, and it is prudent to consider an intermediate step, which is necessary for the accumulation of experience in the design and operation of CF. We refer here to the need for research on the initial use of LNG. Table 2 presents the main characteristics of fuels, which will be required for a conceptual assessment against the background of traditional fuel for gas turbine engines: aviation kerosene TS-1.

**Table 2.** Characteristics of fuels for gas turbine engines.

| Fuel | TS-1 (Kerosene) | Liquefied Methane—$CH_4$ | Liquefied Hydrogen $H_2$ |
|---|---|---|---|
| Density $\rho$, t/m$^3$ | 0.82 | 0.49 | 0.07 |
| Mass calorific value $q_m$, MJ/kg | 42.8 | 50 | 120 |
| Volumetric calorific value $q_v$, MJ/L | 35.1 | 24.5 | 8.4 |
| Boiling temperature $T_b$,°C | 180 | –253 | –162 |

One of the most successful attempts in the direction of the use of CF was made in the USSR under the leadership of designer A.A. Tupolev in the 1980s. At this time, research

began on the bench, ground, and flight tests of the liquefied hydrogen fuel system to test the performance of such new systems and ensure their safe operation. To implement this program at minimal cost, the serial Tupolev Tu-154 aircraft with NK-8-2 engines were converted into the Tupolev Tu-155 flying laboratory. One engine was replaced with an experimental Kuznetsov NK-88 that ran on hydrogen. The developers faced a number of new complex problems in the transition to CF. The first was associated with low temperatures, and the second was associated with the large dimensions of CF tanks (CFT).

CFT must be kept below $-253\ ^{\circ}$C to prevent boiling and requires effective thermal insulation. Given the possible high pressure inside CFT, the tanks must have either a spherical or cylindrical shape to reduce weight. According to the volumetric heat of combustion, kerosene correlates with liquefied hydrogen at 35.1/8.4 = 4.18 and with LNG at 35.1/24.5 = 1.43 (Table 2). This means that when implementing CF with energy efficiency at the same technical level as with kerosene and with the same flight performance characteristics of UAV, this will require a similar increase in the volume of CFT, and as a result, the dimensions of the aircraft. An increase in aerodynamic drag as a result of this, like a snowball, will lead to an increase in fuel consumption and the need to increase engine thrust; the mass of the aircraft will increase, which will also lead to an increase in the mass of the structure. In this regard, on the Tupolev Tu-154, to maintain an aerodynamic shape, CFTs were placed in the rear fuselage compartment, which naturally could "eat" half of the passenger capacity of the aircraft.

Since 1988, this aircraft has made five flights. When the developers faced the difficulties of using liquefied hydrogen, it became obvious that to gain experience, a smoother transition from high-boiling fuel to low-boiling fuel was simply necessary. At this intermediate step, the Tupolev Tu-155 was modified for flights using the more affordable and denser CF: LNG, ranging from $CH_4$ to $C_5H_{12}$ pentane). An important argument for such a transition was not only that LNG during transportation requires a storage temperature below $-162\ ^{\circ}$C, but LNG is much cheaper than not only liquefied hydrogen and also almost two times cheaper than kerosene. LNG is less flammable than hydrogen, and by that time, sufficient experience was accumulated in maintaining such fuel in safe conditions. Since 1989, LNG has made 90 successful test flights. All of them showed that fuel consumption is reduced by almost 15% compared to kerosene (this value corresponds to the ratio of the mass heat capacities of LNG and kerosene), and such a "cryogenic aircraft" can become more economically profitable.

In [11], a fuel-economic analysis of the Tupolev Tu-206 aircraft project was performed, the basic model of which is the existing Tupolev Tu-204 aircraft, re-equipped with three Kuznetsov NK-89 with a total thrust of 320 kN, capable of operating with two separate fuel systems: existing standard on kerosene and new cryogenic for LNG. In this work, it was shown that despite the increase in the mass of the new aircraft by 13%, there is an annual economic effect due to the reduction in LNG costs equal to 1.8% of the cost of the entire Tupolev Tu-204 ($46,000,000). When operating passenger aircraft for 20–30 years, the transition to LNG can provide airlines with significant cost savings, reduce ticket prices for passengers, and, most importantly, reduce the environmental impact on the environment [13].

The most challenging factor in the use of CF for UAVs, which are relatively small in size, is the placement and design of fuel tanks [5–8], the volume of which, taking into account their specific shape, increases significantly compared to traditional hydrocarbon fuels. Proceeding from this, the problem arises of justifying the use and optimal placement of CF fuel tanks on board the UAV [14].

The lack of complete information complicates the design process and requires a conceptual approach with sufficiently simple and reliable results. The most important parameter of any aircraft is its takeoff weight. In most cases of evaluating the use of CF for UAVs at the conceptual level, the listed works do not consider the relation with the take-off mass and do not evaluate this influence, which plays a primary role in deciding on further design.

In the proposed work, based on the analysis of the sensitivity of the takeoff mass to design changes, the conceptual possibility of the rationality of using CF for UAVs is ana-lyzed. The study is conducted on a specific example: the currently widely used MQ-9 Reaper drone with a take-off weight of up to 5 tons, developed by General Atomics Aeronautical Systems (USA). It is equipped with a turboprop engine that allows it to reach speeds of more than 400 km/h and reach an altitude of more than 13.2 km while in flight in the air for up to 25 h. The authors' numerical studies evaluate the possibility of implementing a concept project to use liquefied hydrogen and LNG in metal and composite fuel tank designs. The MQ-9 Reaper UAV is used as a base UAV, against which the rationale of using CF is considered according to the criterion of the minimum takeoff weight.

## 3. Methodology

A reduction in costs and harmful effects on climate and the environment remain relevant for today's rapidly developing direction in aviation: UAV. For any aircraft, including UAV, takeoff weight and, in particular, maximum weight is one of the leading design parameters. In aircraft design, knowledge of the maximum mass is necessary for many reasons. It is taken into account when assessing the design loads, which largely determine the structure's mass when assessing the thrust-to-weight ratio of an aircraft, when choosing the parameters of a power plant, and when determining fuel costs. At the conceptual design stage, up to 80% of all significant technical decisions are made. At the same time, the developers consider and sort through a large number of different options in a fairly limited period of time, so it is desirable to have an "easy" in terms of labor intensity and a sufficiently proven approach to confidently and quickly attain a more effective design direction. In this regard, we used a methodology in this study based on existing (basic) projects and a change in technical requirements.

Several studies [15–22] considered the use of aircraft takeoff mass growth factors due to initial changes to solve such problems. It was shown in [23] that this approach is essentially sensitivity analysis in its simplest form. Sensitivity or sensitivity coefficient is usually understood as the ratio of the change in the final value of the parameter to the initial value. This technique is based on the assessment of sensitivity coefficients of the base UAV to new design changes.

The take-off mass of the aircraft is represented as the sum of the main functional components:

$$m_{TO} = \sum_{i=1}^{4} m_i = m_{str} + m_{p.p} + m_{fuel.s} + m_{targ}, \tag{1}$$

where $m_{str}$ is the mass of the structure (subsystem that combines all functional components into a single whole and ensures the placement and safety of the payload); $m_{p.p}$ is the mass of the power plant (subsystem that provides the creation of thrust: engines, nacelles, frame with attachment points, etc.; we will accept the following dependence of this subsystem on the mass of engines: $m_{p.p} = k_{p.p} \cdot m_{eng}$, where $m_{eng}$ is the mass of the engines, $k_{p.p} \approx 1.1$, $m_{fuel.s}$ is the mass of the subsystem that provides power to the power plant during a given flight time (fuel and devices for its placement and supply, the mass of which is 10–15% of the fuel mass, $m_{targ}$ is the target load mass: subsystem that includes masses associated with the purpose of the aircraft, here this is the commercial (payload) load, crew, payload equipment and equipment that ensures reliable and safe flight).

In general, the fourth term on the right side of Equation (1), $m_{targ}$, is determined mainly by the design task of the aircraft, and the first three terms are directly dependent on $m_{TO}$. To reduce this influence, relative masses are used to estimate the takeoff mass of the initial approximation: $\overline{m}_i = m_i / m_{TO}$, so,

$$m_{TO} = m_{targ} / (1 - \overline{m}_{str} - \overline{m}_{p.p} - \overline{m}_{fuel.s}), \tag{2}$$

New solutions will be implemented through the main parameters of the aircraft and reflected in the $m_{TO}$ of the basic project. The transition to a new project from an already existing (basic) one will require partial changes in the mass of the functional elements of

the system, which will then develop into a general (final) change in the group of the aircraft as a system formed by the interconnection of these elements. Changes in indicators of technical excellence of the primary project through changes $\overline{m}_{str}$, $\overline{m}_{p.p}$, $\overline{m}_{fuel.s}$ will lead to a shift in takeoff mass $m_{TO}$, which can be considered a continuous function of $n$ of variable parameters $q_j$ (weight, aerodynamic, economic, etc.). We express the total mass differential in terms of these parameters and taking into account finite but small changes $q_j$, this transition can be represented as:

$$dm_{TO} = \sum_{j=1}^{n} \frac{\partial m_{TO}}{\partial q_j} dq_j \rightarrow \Delta m_{TO} = \sum_{j=1}^{n} \frac{\partial m_{TO}}{\partial q_j} \Delta q_j. \tag{3}$$

As such, any changes in the takeoff weight design can be evaluated, in particular, due to the initial changes in the corresponding functional weight $\Delta m_{i0}$ ($i = 1-4$, and the index "0" will mean the initial change in mass), then:

$$\Delta m_{TO} = \frac{\partial m_{TO}}{\partial m_j} \Delta m_{i0} = \mu_{mi} \Delta m_{i0}. \tag{4}$$

where $\mu_{mi}$ is the mass sensitivity coefficient (MSC) of the aircraft. Passing from infinitesimal quantities to finite increments, we get:

$$\mu_{mi} = \frac{\partial m_{TO}}{\partial m_j} \approx \frac{\Delta m_{TO}}{\Delta m_{i0}}. \tag{5}$$

Without dwelling on the procedure for obtaining MSC, but referring the reader to [21,23], we present the final value of this coefficient for the case of an initial mass change in the $i$-th functional part of the aircraft while maintaining all the aircraft performance characteristics:

$$\mu_{mi} = 1 / \left[ \overline{m}_{targ} - \Delta \overline{m}_{i0} + \left( \overline{m}_{p.p} + \overline{m}_{fuel.s} \right) C_{Dfus} / C_D \right], \tag{6}$$

where $\Delta \overline{m}_{i0} = \Delta m_{i0}/m$ (to simplify the problem, this relative mass, which introduces relatively small corrections, will not be considered further); $C_{D\ fus}$ and $C_D$ are the drag coefficients of the fuselage and the entire aircraft. The last term with these coefficients in (6) appeared for the case when, while maintaining the payload mass, the fuselage was taken unchanged [21]. When changing the size of the fuselage, this term will no longer exist.

In the final form, the formulas for estimating the masses in the new project will be as follows:

$$\Delta m_{TO} = \mu_m \sum_{i=1}^{4} \Delta m_{j0}, \tag{7}$$

$$\Delta m_i = \Delta m_{i0} + \overline{m}_i \Delta m_{TO}, \ for \ i = 1\text{–}3, \tag{8}$$

$$\Delta m_{targ} = \Delta m_{targ0}, \ for \ i = 4. \tag{9}$$

## 4. Research Results

We consider at the conceptual level the possibility of transferring the UAV to CF. A similar study was carried out for Blended Wing Body aircraft [24]. For numerical studies, we will consider MQ-9 Reaper, a medium-altitude aircraft with long flight duration, as the base. It is currently the main strike UAV of the USA Air Force. Figure 1 shows three projections of the MQ-9 Reaper with the main average dimensions [25].

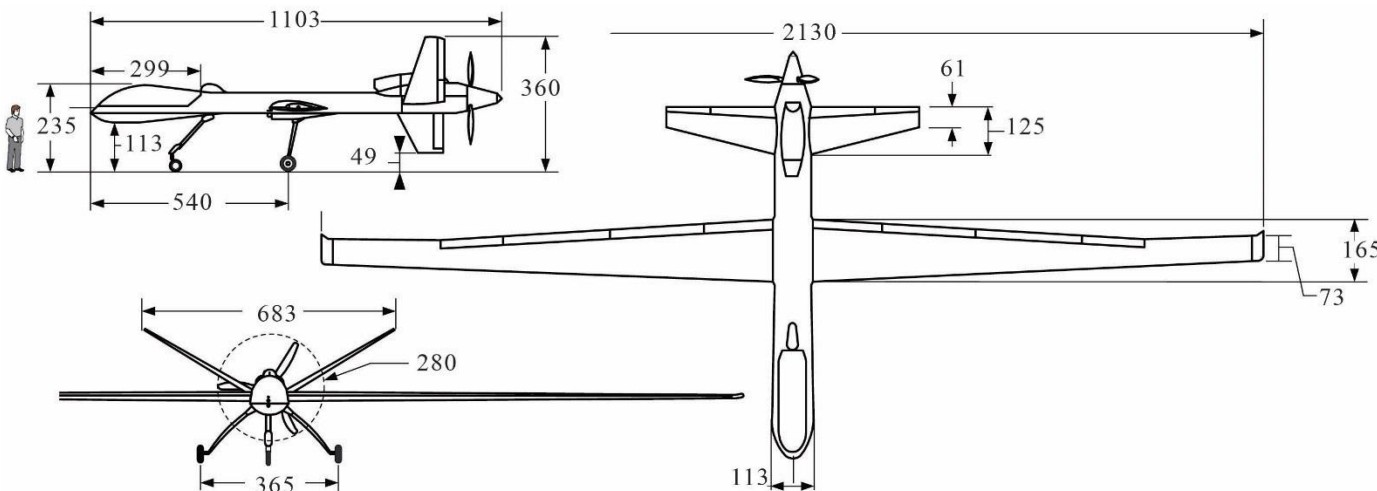

**Figure 1.** MQ-9 Reaper (in cm) [25].

To calculate MSC, it is necessary to know the functional masses of the base aircraft. Honeywell TPE331-10T turboprop engine with an output power of 712 kW used on this UAV has a mass of 175 kg and specific fuel consumption in terms of power $C_N$ = 325 g/(kWh). In addition to the engine, the mass of the power plant will include a gearbox, propeller, motors, and automation. The heaviest part after the engine is the gearbox. Figure 2 [19] shows the statistical dependence of the mass of the gearbox on the transmitted power. Based on this, we accept the following dependence: $m_{p \cdot p}$ = 1.7, $m_{eng}$ = 300 kg, so, $\overline{m}_{eng}$ = 0.062.

To calculate MSC, it is necessary to know the functional masses of the base aircraft. To estimate the relative mass of the structure, we use the data of [26]. Figure 3 shows the dependency graph $m_{str}$ on the specific wing load $p$ for a design with extensive use of composite materials. For a basic aircraft whose wing area is $S$ = 25.4 m$^2$, $p$ = 187.4 kg/m$^2$, we obtain $m_{str}$ = 0.29 and $m_{str}$ = 1389 kg.

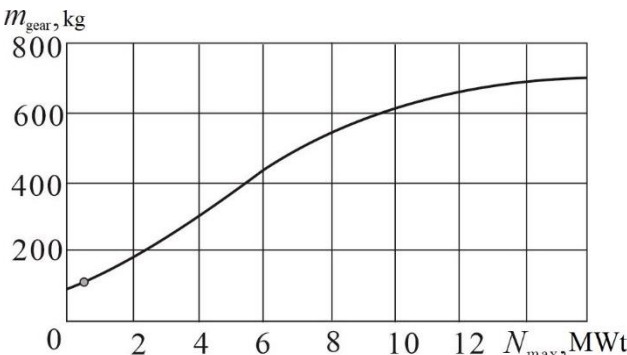

**Figure 2.** Dependence of mass of gearbox on the transmitted power [27].

At the maximum mass of fuel $m_{fuel}$ = 1800 kg, we assume that the mass of the entire fuel system is $m_{fuel \cdot s} = k_{fuel \cdot s} \cdot m_{fuel}$. If $k_{fuel \cdot s}$ = 1.055, we obtain $m_{fuel \cdot s}$ = 1900 kg, so, $m_{fuel \cdot s}$ = 1900/4 750 = 0.4. The payload mass for the variant with the maximum fuel mass will be $m_{p \cdot l} = m_{T.} - m_{emp} - m_{fuel}$ = 4760 − 2223 − 1800 = 737 kg.

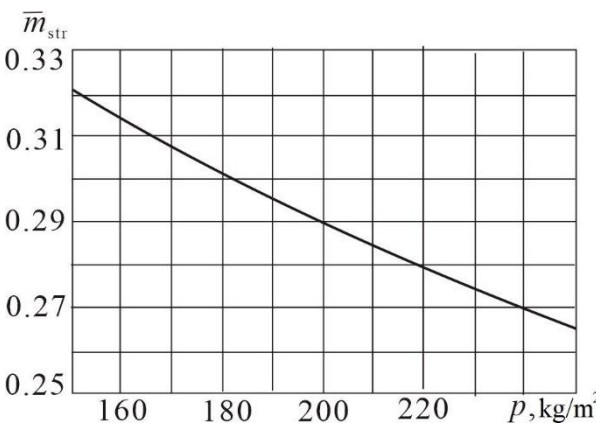

**Figure 3.** Dependence of relative mass of structure on the specific load on a wing [28].

From the knowledge of the empty mass of the aircraft, which includes the mass of the structure, power plant, dry part of the fuel system, and non-removable equipment, it is possible to find the mass of equipment and equipment $m_{eq}$, which, as we have already noted, is considered to be the mass independent of the takeoff:

$$m_{eq} = m_{emp} - (m_{str} + m_{p \cdot p} + m_{fuel \cdot s. \ emp}) = 2223 - (1389 + 300 + 100) = 434 \text{ kg.}$$

Thus, the mass of the target load will be:

$$m_{targ} = m_{eq} + m_{p \cdot l} = 434 + 737 = 1171 \text{ kg.}$$

Thus, according to the results of the preliminary analysis, the base aircraft has the following relative functional masses: $m_{str} = 0.3$; $m_{p \cdot p} = 0.06$; $m_{fuel \cdot s} = 0.4$; $m_{targ} = 0.24$.

MSC to the initial change in mass, provided that the aerodynamic parameters of the fuselage and the accepted ratio are preserved $C_{D \ fus}/C_D = 0.3$ according to (6) will be: $\mu_m = 1/(0.24 + (0.4 + 0.06) \times 0.3) = 2.65$.

In the event of a change in the size of the fuselage; this may occur when it is re-arranged for large MSC, $\mu_m = 1/0.24 = 4.2$.

When switching to CF, we will assume that the mass of turboprop engine will remain the same, and the traction characteristics will change in proportion to the mass heat of combustion. Then the values of the mass of CF will be:

- For LNG: $m_{fuel \ LNG} = q_{m \ LNG}/q_m \ m_{fuel} = 50/42.8 \times 1.8 = 1.54$ t;
- For hydrogen: $m_{fuel \ liquefied \ hydrogen} = q_{m \ liquefied \ hydrogen}/q_m \ m_{fuel} = 120/42.8 \times 1.8 = 0.64$ t.

Thus, the initial change in the mass of fuel will be:

- For LNG: $\Delta m_{fuel \ LNG \ 0} = -0.26$ t;
- For hydrogen: $\Delta m_{fuel \ liquefied \ hydrogen \ 0} = -1.16$ t.

Having calculated the required volumes of CFT and increasing them by 15% (taking into account the free volumes required for refueling and for fuel fittings), we obtain, respectively: $W_{LNG} = 3.6 \text{ m}^3$, $W_{liquefied \ hydrogen} = 10.5 \text{ m}^3$.

We will evaluate the possibilities of installing CFT on UAV. For cylindrical CFT with radius $r$, we use spherical bottoms with radius $R$. To ensure equal strength of MSC and minimize its mass; we take $R = 2r$. In this case, the removal of the bottom will be $h = (2 - (3)^{0.5}) \ r = 0.27 \ r$, and the volume that forms the bottom (spherical segment with radius $R$, height $h$, and base radius $r$):

$$W_{bott} = \pi \times h^2(R - h/3) = \pi \times 0.14 \ r^2 \times (2 \ r - 0.09 \ r) = \pi \times 0.27 \ r^3. \tag{10}$$

As a result, the volume of the entire tank can be represented as:

$$W_{CFT} = W_{CFTcyl} + 2W_{CFTbott} = \pi r^2 \left( l_{CFT} - 0.54\,r \right). \tag{11}$$

Figure 4 shows the dependencies for the design dimensions of CFT: lengths and radius for the given volumes.

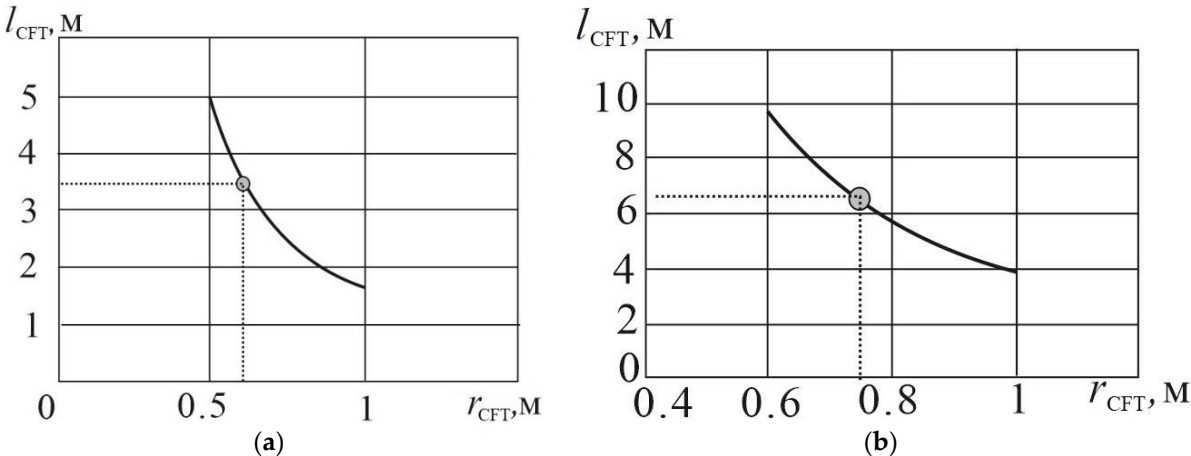

(**a**) (**b**)

**Figure 4.** Dependence of the length of CFT on the radius: (**a**)—for LNG (volume 3.6 m$^3$); (**b**)—for liquefied hydrogen (volume 10.5 m$^3$) [25].

According to the layout conditions of CFT on the fuselage, we chose the following values: for LNG $r_{CFT} = 0.55$ m, $l_{CFT} = 4$ m; for liquefied hydrogen $r_{CFT} = 0.75$ m, $l_{CFT\,F} = 6.4$ m.

We will link CFT with the base UAV. Given the civilian use of the aircraft, it can be assumed that the forward fuselage can be changed. In this case, the midship section with the same dimensions as the nose of the MQ-9 Reaper is extended to the attachment point of the wing with the fuselage (Figure 5) when installing CFT with LNG (Figure 6). Such an arrangement of the tank will ensure the preservation of the power scheme for attaching the wing to the fuselage.

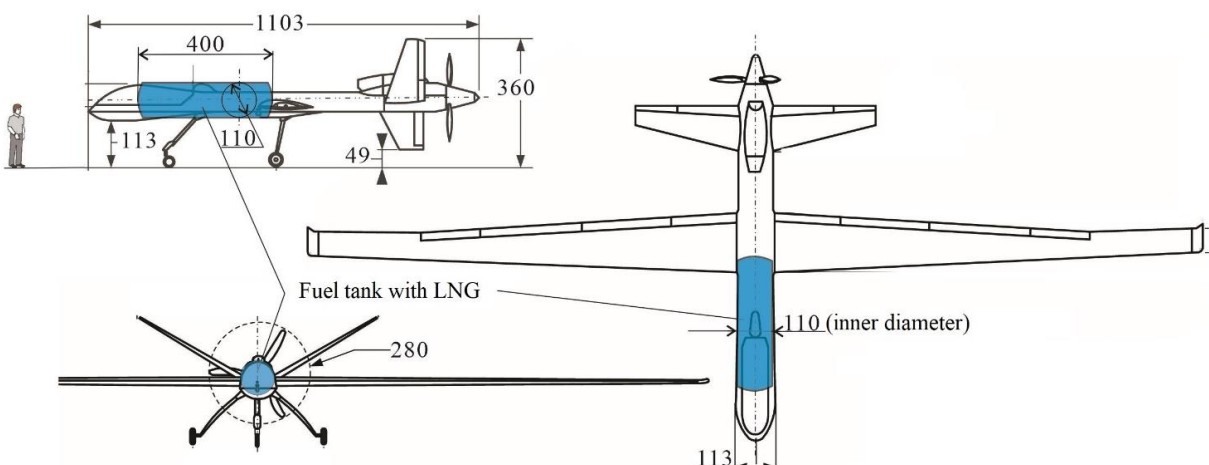

**Figure 5.** Initial linkage of CFT with LNG with a basic civilian UAV (the size in cm).

For CFT with hydrogen, an increase in the diameter of the fuselage will be required (Figure 7). In addition, CFT in a single-tank version will break the center section of the wing and will need power frames for attaching the wing consoles and additional niches for the main landing gear. All this will cause an increase in aerodynamic drag and an increase in

the mass of the structure. In addition, the propeller will be heavily shaded by the fuselage with a large midsection, and it will be necessary to transfer the power plant from the rear fuselage to the nose (to use a pulling rather than a pushing propeller group). In this option, it is more rational to make two CFTs. The length of each tank with an inner diameter of 1.5 m is 3.4 m.

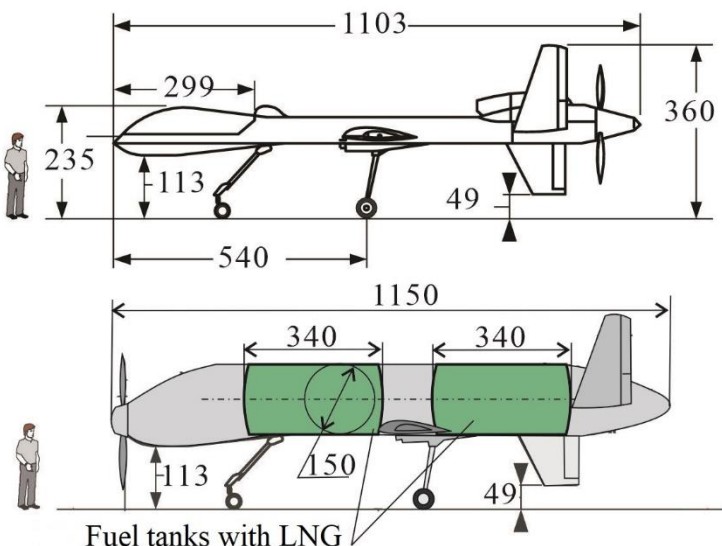

**Figure 6.** Comparison of a basic UAV with a two-tank version on LNG with a reconfiguration of the power plant (the size in cm).

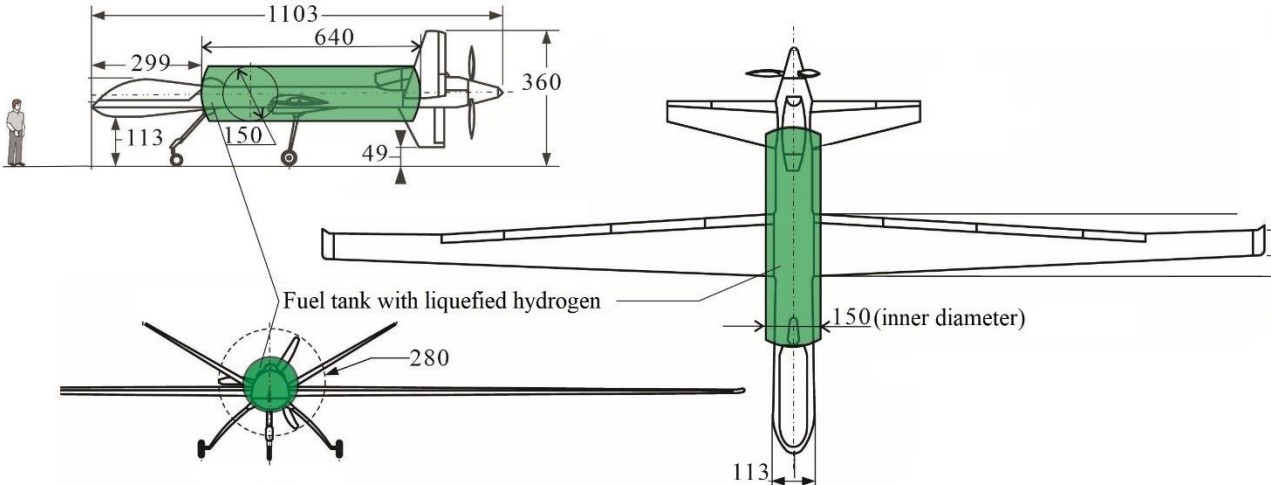

**Figure 7.** Initial linkage of CFT with liquefied hydrogen on the base UAV in a single-tank version (the size in cm).

It is necessary to add the mass of the tank with thermal protection to the fuel system. In this case, we will assume that the previously existing tanks for kerosene with a system for supplying it to the engine compensate for the mass of the supply system for the cryogenic component.

For a design estimate of the mass of CFT without delving into its design, we will use the results of [26], in which the mass of a thin-walled cylindrical structure of a circular cross section, calculated for excess internal pressure *p*, was determined using the forth theory of strength according to the following dependencies:

- For a structure made of isotropic material (metal):

$$m_{CFT\ 0} = 3^{1/2}\ pW/\overline{\sigma}_{MET},\tag{12}$$

- For a structure made of composite material:

$$m_{CFT\ 0} = 3\ pW/\overline{\sigma}_{comp\ mat},\tag{13}$$

where $\overline{\sigma}$ is the specific strength of the material, representing the ratio of allowable stresses to the density of the material: $\overline{\sigma} = \sigma_B/\eta/\eta_p/\rho$, where $\eta$ is the overall safety factor, and $\eta_p$ is the safety factor associated with work under excessive pressure. For composite material, an additional safety factor is introduced: $\eta_{comp\ mat}$.

When calculating the initial mass costs for fuel and for the power plant, associated with an increase in aerodynamic resistance, for a cruising flight, the following relation can be taken:

$$\Delta D_0 = \Delta T = T/(m_{eng.s} + m_{fuel.s})(\Delta m_{eng.s\,0} + \Delta m_{fuel.s\,0}),\tag{14}$$

So, it follows:

$$\Delta D_0 = g/K(\overline{m}_{eng.s} + \overline{m}_{fuel.s})(\Delta m_{eng.s\,0} + \Delta m_{fuel.s\,0}).\tag{15}$$

And then, according to the definition, CFT in terms of aerodynamic drag is:

$$\mu_D = \Delta m_{TO}/\Delta D_0 = \mu_m K(\overline{m}_{eng.s} + \overline{m}_{fuel.s})/g.\tag{16}$$

To calculate the change in the force of aerodynamic drag due to a change in the diameter of the midsection of the fuselage ($d_{fus} \Rightarrow d_{fus\ New}$), we assume that the resistance of the fuselage is proportional to its midsection area: $S_{fus}$, so:

$$D_{fus\ New}/D_{fus} = S_{fus\ New}/S_{fus} = (d_{fus\ New}/d_{fus})^2.\tag{17}$$

The average value of the aerodynamic drag of the base UAV is considered at a half the fuel consumption:

$$D_{fus} = 0.3D = 0.3(m_{TO} - 0.5m_{fuel})g/K.\tag{18}$$

Then, the change in the drag force of the fuselage for new versions of UAV, assuming the invariance of the drag coefficient, can be calculated from the following relation:

$$\Delta D_{fus} = D_{fus}\ ((d_{fus\ New}/d_{us})^2 - 1),\tag{19}$$

where $d_{fus\ New}$ is the diameter of the fuselage midsection of the new UAV, which is determined by the dimensions of CFT, taking into account its heat-shielding coating.

## 5. Discussion

We analyze the values of the masses for two variants of fuel tanks: aluminum alloys and composite materials. The following parameter values were taken for calculation: $\eta = 1.5$; $\eta_p = 3.5$; $\eta_{comp\ mat} = 1.5$; $\sigma_{Balum\ all} = 600$ MPa; $\sigma_{Bcomp\ mat} = 1500$ MPa; $\rho_{alum\ all} = 2.7$ t/m$^3$; $\rho_{comp\ mat} = 1.5$ t/m$^3$; $p = 1$ MPa. For thermal insulation of CFT, a polyurethane foam coating with a purge system with a total density of the system was considered $\rho_{thermal\ insulation} = 100$ kg/m$^3$ and the thickness of such a heat-protective coating (HPC): for CFT with LNG $\delta_{HPC} = 0.1$ m, for CFT with liquefied hydrogen $\delta_{HPC} = 0.2$ m. The following parameters were adopted for the base UAV: fuselage midsection diameter $d_{fus} = 1.13$ m; $K = 25$.

The resistance of the fuselage of the base UAV in the cruising flight mode according to (18) and the accepted values of the parameters will be $D_{fus} = 454$ N.

In Table 3 are the results of a conceptual analysis of the possibility of transforming a basic UAV of MQ-9 Reaper into CF.

**Table 3.** Estimation of mass characteristics of LNG and liquefied hydrogen UAV with two types of fuel tanks.

| Type of Fuel for UAV | LNG | | Liquefied Hydrogen | |
|:---:|:---:|:---:|:---:|:---:|
| CFT Material | Aluminum Alloys | Composite Materials | Aluminum Alloys | Composite Materials |
| MSC $\mu_m$ | 2.65 | 2.65 | 4.2 | 4.2 |
| $\Delta m_{\text{fuel 0}}$, t | −0.26 | −0.26 | −1.16 | −1.16 |
| $\Delta m_{\text{CFT 0}}$, t | 0.085 | 0.047 | 0.46 | 0.14 |
| $\Delta m_{\text{HPC 0}}$, t | 0.16 | 0.16 | 0.67 | 0.67 |
| $\Delta m_{\text{TO 1}} = \mu_m(\Delta m_{\text{fuel 0}} + \Delta m_{\text{CFT 0}} + \Delta m_{\text{HPC 0}})$, t | −0.039 | −0.140 | −0.126 | −1.47 |
| $d_{\text{fus}}$, m | 1.3 | 1.3 | 1.9 | 1.9 |
| $\Delta D_{\text{fus 0}}$, N | 146 | 146 | 829 | 829 |
| $\Delta m_{\text{aer 0}} = \Delta m_{\text{fuel 0}} + \Delta m_{\text{p.p 0}}$, t | 0.172 | 0.172 | 0.973 | 0.973 |
| $\Delta m_{\text{TO aer}} = \mu_m \Delta m_{\text{aer 0}}$, t | 0.456 | 0.456 | 4.087 | 4.087 |
| $\Delta m_{\text{TO}} = \Delta m_{\text{TO 1}} + \Delta m_{\text{TO aer}}$, t | 0.417 | 0.316 | 3.961 | 2.618 |
| $\Delta m_{\text{TO}}/m_{\text{TO}}$ 100% | 8.76 | 6.64 | 83.2 | 54.8 |

New values of functional masses are determined by formula (8).

From the analysis of the obtained values of $\Delta m_{\text{TO}}$, given in the last row of Table 3, it is quite obvious that for such a class of UAV (about 5 t of takeoff weight), the use of liquefied hydrogen is irrational. This is due to the increase in the transverse size of the fuselage to accommodate CFT, which significantly increases aerodynamic drag and the corresponding fuel consumption. When using LNG, the takeoff weight increases by about 6–8% and the final conclusion on the rationale for the use of such fuel should be made taking into account detailed economic analysis of the design situation and with a more accurate assessment of all design parameters, including data on fuel, operating costs, specifics of UAV operation, etc.

## 6. Conclusions

The main achievement of this work is the development of a conceptual methodology for weighting due to design changes in the base UAV (in this case, the transition from hydrocarbon fuel to CF).

The prospect of using liquefied gases (methane and hydrogen) requires a preliminary assessment of the impact of changes in the design of UAVs. MQ-9 Reaper drones were chosen as the objects of study. The calculation results showed that for UAVs of this class when using LNG, the take-off weight increases by about 6–8%, and the conclusion on the justification for the use of such fuel should be made taking into account a detailed economic and environmental analysis of the design situation and with a more accurate assessment of all design parameters, including data on fuel, operating costs, specifics of UAV operation, etc. For the considered UAV, the use of liquefied hydrogen remains unjustified.

**Author Contributions:** Conceptualization, A.K.; methodology, V.G.; software, A.T.; validation, V.G. and A.T.; formal analysis, A.K.; investigation, A.T.; resources, V.G.; data curation, A.K.; writing—original draft preparation, A.K. and V.G.; writing—review and editing, A.K., V.G. and A.T.; visualization, V.G.; supervision, A.T.; project administration, A.T.; funding acquisition, A.K. All authors have read and agreed to the published version of the manuscript.

**Funding:** This research received no external funding.

**Institutional Review Board Statement:** Not applicable.

**Informed Consent Statement:** Not applicable.

**Data Availability Statement:** Not applicable.

**Conflicts of Interest:** The authors declare no conflict of interest.

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
