# Peer review of "Conceptual Assessment of the Possibility of Using Cryogenic Fuel on Unmanned Aerial Vehicles"

_drones, doi:10.3390/drones6080217_

Round 1

Reviewer 1 Report

This study is devoted to the creation of modern unmanned aerial vehicles (UAV), the most efficient from economic and environmental points of view. The topic is interesting and exhibits some novelty. However, the following major concerns should be addressed.

1. It is highly suggested to improve the quality of some figures. These figures can be easily improved for changing their quality, e.g., the eps files can be used in this work.

2. Related works are only listing the existing work without critically analyzing the pitfalls in existing work.

3. Related to point 2, motivations are not clear. In addition, it is difficult to understand the actual contribution of the paper. Key contribution of the paper should be mentioned in bullets form at the end of introduction section.

4. At the end of the intro, I would suggest adding a clear paper organization.

5. The authors mentioned that the study is devoted to the creation of modern unmanned aerial vehicles (UAV), the most efficient from economic and environmental points of view. However, it generates some doubts to me. Specifically, the paper in its current form is very unidirectional; without any comparisons of technologies and futuristic challenges and dispositions, also it lacks to address any alternative and more competitive approach towards the challenges under discussion. I would strongly suggest adding a multi-channel outlay to the research by adding socio-economic dimension or potential use cases alongside technical details.

6. The specific problems studied in this paper are not clear. I would suggest adding a clear research question based on the discussion and argumentation.

Author Response

Reviewer 1

This study is devoted to the creation of modern unmanned aerial vehicles (UAV), the most efficient from economic and environmental points of view. The topic is interesting and exhibits some novelty. However, the following major concerns should be addressed.

  1. It is highly suggested to improve the quality of some figures. These figures can be easily improved for changing their quality, e.g., the eps files can be used in this work.

We send you the source files in CorelDRAW by email, they have the good quality.

  1. Related works are only listing the existing work without critically analyzing the pitfalls in existing work.

The analysis of cited papers is done.

  1. Related to point 2, motivations are not clear. In addition, it is difficult to understand the actual contribution of the paper. Key contribution of the paper should be mentioned in bullets form at the end of introduction section.

At the end of the Introduction, which has been reformatted, an addition has been made.

  1. At the end of the intro, I would suggest adding a clear paper organization.

A clear paper organization is added

  1. The authors mentioned that the study is devoted to the creation of modern unmanned aerial vehicles (UAV), the most efficient from economic and environmental points of view. However, it generates some doubts to me. Specifically, the paper in its current form is very unidirectional; without any comparisons of technologies and futuristic challenges and dispositions, also it lacks to address any alternative and more competitive approach towards the challenges under discussion. I would strongly suggest adding a multi-channel outlay to the research by adding socio-economic dimension or potential use cases alongside technical details.

In the work, the main issue related to the layout is solved: the possibility of placing the cryogenic fuel tanks on the UAV. Consideration of other aspects is no longer within the scope of the article, but it is expected to be considered in future publications.

  1. The specific problems studied in this paper are not clear. I would suggest adding a clear research question based on the discussion and argumentation.

A clear research question based on the discussion and argumentation is added.

Reviewer 2 Report

A preliminary analysis of the effects of alterations to the UAV's design is necessary in light of the possibility of employing liquefied gases (methane and hydrogen).

Overall, the paper is well written and easy to follow. However, I think that the following points should be addressed prior acceptance.

I think that there is a methodological (in writing) issue with this paper. There is no statement clearly showing what is the contribution/finding of this work in any part of the paper (although it can be concluded easily).

The whole paper can be better structered with more sections for the sake of clarety

I would also suggest to include a paragraph showing the overall organization of the paper at the end of the introduction

I do not understand why the cited works are all outdated except the authors own cited work!!

Besides the values of the masses for two variants of fuel tanks, can't we also consider the energy consumption model and other charactiristics? 

Author Response

Reviewer 2

A preliminary analysis of the effects of alterations to the UAV's design is necessary in light of the possibility of employing liquefied gases (methane and hydrogen).

Overall, the paper is well written and easy to follow. However, I think that the following points should be addressed prior acceptance.

I think that there is a methodological (in writing) issue with this paper. There is no statement clearly showing what is the contribution/finding of this work in any part of the paper (although it can be concluded easily).

The whole paper can be better structered with more sections for the sake of clarety.

The sections are added.

I would also suggest to include a paragraph showing the overall organization of the paper at the end of the introduction.

A paragraph is added.

I do not understand why the cited works are all outdated except the authors own cited work!!

The references of the other authors’ works are added.

Besides the values of the masses for two variants of fuel tanks, can't we also consider the energy consumption model and other charactiristics?

The remark is very fair, this model will be considered in future works when using cryogenic fuel on UAV of a larger mass when comparing options with fuel cells and gas turbine engines.

Reviewer 3 Report

The subject of the article is current and interesting. However, after reading this article carefully, the following points were formulated.

- line 51: "CF" - there is no explanation of this abbreviation here; only further in the text is "cryogenic fuel";

- lines 115-189: this text does not contain a methodology but is rather a description of the problem in general with the included case study for the Tu-155 aircraft; therefore it should be placed before the "2. Methodology" section; I propose to add a section "Literature review" for this purpose and insert this text there and do a solid literature review;

- lines 216-220: what "variable parameters" is written about? there is no clarification on this issue;

- Fig. 2, Fig. 3,Fig. s4: the source of these figures is missing;

- lines 375-380: this text is not a summary, but general characteristics of the UAV;

- Fig. 7: there is no reference to this in Fig. 7 in the text;

- the results of the calculations for the case study Honeywell TP331-10T (UAV MQ-9 Reaper) are presented. But I did not find an indication of the calculations for the Helios UAV.

- there is no clear indication of the scientific contribution of the presented results to the state of the art;

Author Response

Reviewer 3

The subject of the article is current and interesting. However, after reading this article carefully, the following points were formulated.

- line 51: "CF" - there is no explanation of this abbreviation here; only further in the text is "cryogenic fuel";

There is the explanation of CF in the Abstract.

- lines 115-189: this text does not contain a methodology but is rather a description of the problem in general with the included case study for the Tu-155 aircraft; therefore it should be placed before the "2. Methodology" section; I propose to add a section "Literature review" for this purpose and insert this text there and do a solid literature review;

The above-mentioned text is placed before the "2. Methodology" section. The Literature Review is expanded and it is included in the Introduction and in the Methodology.

- lines 216-220: what "variable parameters" is written about? there is no clarification on this issue;

The explanations are added.

- Fig. 2, Fig. 3,Fig. s4: the source of these figures is missing;

The sources are added.

- lines 375-380: this text is not a summary, but general characteristics of the UAV;

Thank you for the comment. It is the general characteristics.

- Fig. 7: there is no reference to this in Fig. 7 in the text;

The sources are added.

- the results of the calculations for the case study Honeywell TP331-10T (UAV MQ-9 Reaper) are presented. But I did not find an indication of the calculations for the Helios UAV;

Due to the lack of necessary data, calculations for the UAV Helios could not be performed.

- there is no clear indication of the scientific contribution of the presented results to the state of the art.

Scientific contribution included in Conclusion.

Round 2

Reviewer 1 Report

I have no further comment

Author Response

Dear reviewer,

Thank you very much for your comments.

Best regards,

Alexey

Reviewer 2 Report

i've no other concers

Author Response

(The authors gave the same response as above.)

Reviewer 3 Report

The Authors partially revised the manuscript. However, the following points still need to be clarified.

- lines 87: "CF" - still in the main text of the manuscript this abbreviation is not explained in the place where the abbreviation is first written; the explanation should be written regardless of the explanation written in the "Abstract" section;

- the "Literature review" section is still not separated from the text; the separation of the literature review between the sections "Introduction" and "Methodology" makes the text unclear;

- there is still text on lines 451-457 that in the section "Conclusion" is unnecessary in the context of the conclusions of the presented research;

- still Fig.7 is not cited in the main text of manuscript;

Author Response

"CF" is now explained in the place where the abbreviation is first written;

the "Literature review" section is now added in the text; 

the text on lines 451-457 from in the section "Conclusion" is moved to "Introduction" section;

the Fig.7 is  cited in the main text of manuscript.